# The association between physical activity intensity and frailty risk among older adults across different age groups and genders: Evidence from four waves of the China Health and Retirement Longitudinal Survey

Di Ma, Yulin Sun◉*, Guoyang Chen, Siwei Hao, Zhenping Jiang, Rui Wang, Shuaipeng Hao

Department of Sports Science, College of Sports & Arts, Hanyang University ERICA Campus, Ansan, South Korea

* mabron606717@gmail.com

## Abstract

"Exercise is the best medicine" is well known, but the optimal dose of physical activity (PA) for males and females across different age groups is still unknown. This study, using data from the four waves of CHARLS, aimed to determine the optimal PA dose that reduces frailty risks among older adults across various age groups and both sexes. We created a frailty index score using 63 health-related variables and used 0.21 as the frailty cut point. Binary logistic regression was used to compare the effect of vigorous, moderate, and light intensity PA under IPAQ criteria on frailty risk. The study found that regardless of whether males or females, the optimal effect of vigorous-intensity PA in reducing the risk of frailty is consistently observed throughout the entire old age career. Moreover, the age groups at which moderate-intensity PA reduces the risk of frailty were from age 70 for males and from age 80 for females. And light-intensity PA had no effect on reducing the risk of frailty. Moderate and vigorous intensity of PA in older adults should be promoted, but guidelines and recommendations must account for optimal associations with PA dose across genders and age groups.

## Introduction

The impact of demographic aging on society has become a global issue that cannot be ignored and poses many challenges in terms of economic development, social security, and health services. According to the predictions of the World Health Organization [1], the proportion of older adults in the global population is expected to increase to 22% by 2050. Meanwhile, China remains one of the countries with the largest population base facing aging issues in the world, with the number of adults over 60 years old expected to reach 402 million by 2040 [2]. This leads to a massive increased demand for health care and security, which places a heavy burden

**Data Availability Statement:** The following data can be downloaded at: http://charls.pku.edu.cn/en/, China Health and Retirement Longitudinal Study.

**Funding:** The author(s) received no specific funding for this work.

**Competing interests:** The authors have declared that no competing interests exist.

on individuals, families, and society. In response to this, it is necessary to shift from the previous concept of a disease-oriented, hospital-based approach towards a direction focused on preventive aging and centered around community or home care [3].

The process of aging is marked by a decrease in functional ability and an increase in susceptibility to illness, disability, and mortality. This is driven by the gradual, lifelong accumulation of molecular and cellular defects [4]. Meanwhile, frailty is one of the key physiological indicators of the individual aging process and is also a common syndrome among older adults. This is manifested by a reduced ability to recover from health stress due to reduced strength, endurance, and physiological function [5]. Although the incidence of frailty gradually increases with age, frailty does not equate to aging. This is explained by the fact that individuals of a given chronological age differ in the degree of progression of the aging process. The age difference in entering a state of increased frailty risk may be related to factors such as different lifestyles [6]. Prolonging the aging process for as long as possible with appropriate preventive measures is defined as healthy aging [7, 8].

Recent studies have shown that active physical activity (PA) is also considered an important preventive measure to prolong the aging process. This prolonged process covers several key aspects, including but not limited to reduced risk of death [9], lower incidence of chronic diseases [10], reduced risk of functional loss [11], improved cognition, and reduced anxiety and depression [12]. The impact of PA heavily relies on its intensity, with vigorous-intensity exercises more effective in enhancing aerobic capacity and cardio protection compared to moderate-intensity PA [13, 14]. Furthermore, the cognitive performance decline in aging manifests as reduced information processing speed and a diminished pool of cognitive resources for processing, storing, retrieving, and transforming information [15]. Physical exercise is largely influenced by the lifestyle of each country, with China particularly influenced by its traditional culture. This influence is particularly evident in the exercise habits of the elderly population. For instance, many older adults in China prefer practicing Tai Chi in parks in the mornings and taking part in square dancing activities in community squares after dinner. The Li et al. study showed that recreational sports with the same intensity as square dancing and walking had an improving effect on the cognitive performance of older adults [16].

Despite numerous studies that have described the health-improving function of PA intensity, more focus has been placed on the effect of PA on a single disease or function. Due to the fact that the complete explanation of the male-female health-survival paradox mechanism has long remained a challenge [17–19], few studies have considered the association between PA intensity and frailty at different age groups for both male and female older adults, including research targeting Chinese older adults. Thus, the present study aimed to examine how various PA intensities in older adults are associated with frailty changes, aiming to provide evidence for preventing frailty via suitable PA doses across different age groups and genders.

## Materials and methods

### Data source and sample selection

The present study utilized data from the China Health and Retirement Longitudinal Survey (CHARLS), which was available through the website of the National Development Institute of Peking University [20]. CHARLS was a nationwide survey of Chinese adults aged 45 years and over that used PPS sampling (probabilities proportional to size) to ensure the representativeness of the data. Multi-stage sampling was conducted using rural, urban, and regional gross domestic product (GDP) per capita as the stratification basis [21]. Over the course of four waves, specifically in 2011, 2013, 2015, and 2018, an extensive survey was conducted encompassing a vast geographical area of 150 counties and 450 communities (villages) spread across

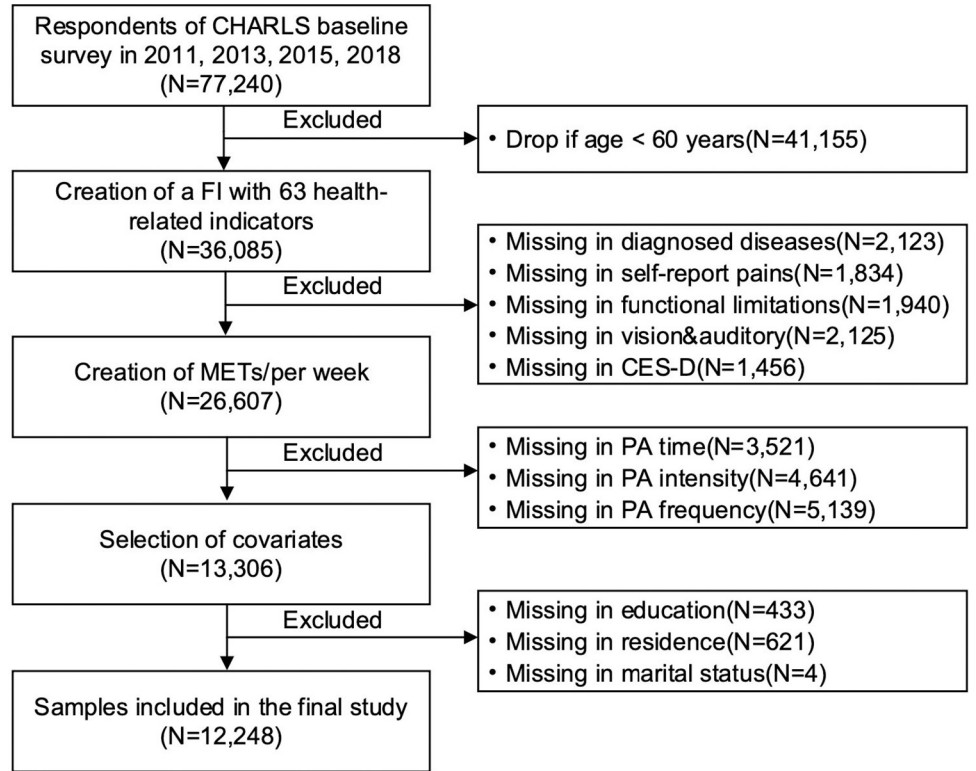

**Fig 1. Flowchart of sample selection.** Note: FI: Frailty Index; METs: Metabolic Equivalent of Task; CES-D: Center for Epidemiologic Studies Depression Scale; PA: Physical Activity.

30 provinces, autonomous regions, and municipalities. The selection of counties and communities had been based on careful considerations to ensure a diverse representation of the population [22]. The Peking University Ethical Review Committee gave the CHARLS their seal of approval (IRB00001052-11015). At the time of participation, each respondent signed the informed consent form, and they were assured of the confidentiality and anonymity of their data.

A total of 77,240 respondents from 4 waves of baseline surveys (2011, 2013, 2015, 2018) were selected from the CHARLS database for this study. Once missing values regarding the frailty index (FI), PA intensity, and covariates were eliminated, the focus of the study was ultimately narrowed down to a valid sample of 12,248 individuals aged 60 years and older (Fig 1). The rigorous data refinement process ensured the quality and reliability of the dataset used for analysis and subsequent research investigations.

### Frailty index

The present study utilized the FI framework created by Rockwood and Mitnitski [23], which calculates the ratio of an individual's deficits to the total number considered. The score ranges from 0 (the lowest degree of frailty) to 1 (the highest degree of frailty). An optimal FI should consist of at least 30 items, covering various health indicators such as chronic conditions, physical or cognitive limitations, and overall health. Each deficiency should be health-related and increase with age while avoiding early saturation. To ensure the index's quality, redundant and duplicate items were minimized, and items with a missing value of more than 5% were discarded [24]. The final FI for this study consisted of 63 items covering the areas of

hospitalization history, diagnosed disease, self-report pains, activities of daily living (ADL), instrumental activities of daily living (IADL), mini-mental state examination (MMSE), center for epidemiologic studies depression scale (CES-D), vision, audition, and dental health. More specifically, each deficit is assigned a score of 0–1 according to the answer to indicate the presence or absence of a deficit. For example, the question "Have you been diagnosed with hypertension by a doctor?" and "No" receives a score of 0. "Yes" receives a score of 1. Accumulate all scores and divide by 63 to get a value of 0–1, which is a FI for continuous type variable.

## Physical activity intensity

The type of PA, duration, and frequency of PA of the respondents were investigated by the CHARLS personnel. The weekly PA intensity of the respondents was quantified based on the criteria of the International Physical Activity Questionnaire Short Forms (IPAQ-SF) [25]. According to the different types of PA, the International Physical Activity Questionnaire categorizes them into three levels: light-intensity PA, moderate-intensity PA, and vigorous-intensity PA. To quantify the energy expenditure of these activities, IPAQ assigns corresponding metabolic equivalent of task (MET) coefficients. Specifically, the coefficient for light-intensity PA is 3.3, the coefficient for moderate-intensity PA is 4.0, and the coefficient for vigorous-intensity PA is 8.0. These coefficients serve as indicators of the relative energy expenditure associated with each intensity level of PA. The calculation formula is:

$$\text{Total MET min/week} = \text{Light } (3.3*\text{min}*\text{days}) + \text{Moderate } (4.0*\text{min}*\text{days}) + \text{Vigorous } (8.0*\text{min}*\text{days}) \quad (1)$$

According to the IPAQ criterion, an activity is classified as 'vigorous-intensity' if an individual engages in 7 or more days of any combination of walking, moderate-intensity, or vigorous-intensity activities, resulting in a minimum total PA of at least 3,000 MET-minutes per week. Activities categorized as 'moderate-intensity' require engaging in 5 or more days of any combination of walking, moderate-intensity, or vigorous-intensity activities, achieving a minimum total PA of at least 600 MET-minutes per week. Activities below 600 MET-minutes per week are classified as 'light-intensity'. Based on these criteria, the respondents were categorized into a vigorous-intensity group, a moderate-intensity group, and a light-intensity group.

## Covariates

The ecological model of behavior suggests that the physical environment can influence participation in PA through a range of mediating and moderating processes. These mediators and moderators have been shown to include sociodemographic, long-term lifestyle habits, and psychological, physical, and environmental factors [26]. In this study, five factors were selected as covariates: marital status, educational background, place of residence, and long-term lifestyle habits of smoking and alcohol consumption. The covariates were used as moderators of grouped randomness in the behavior of older adults, which influenced whether or not they engaged in PA.

## Statistical analysis

In descriptive statistics, continuous variables were expressed as means and standard deviations, and categorical variables as percentages. The characteristics of respondents in different subgroups were tested using one-way ANOVA or chi-square test. Cronbach's alpha was employed to assess the internal consistency of the 63 FI items and the IPAQ according to PA intensity categories (light, moderate, and vigorous groups). A value exceeding 0.70 signifies an

acceptable level of internal consistency [27]. Frequency distribution plot was utilized to illustrate the disparities in FI between males and females within the sample. Logistic regression was utilized to analyze the relationship between FI and PA intensity in older adults. Respondents were categorized by age range into five subgroups: "60–64 years", "65–69 years", "70–74 years", "75–79 years", and "80 years and older". A FI value of 0.21 was defined as the cut point for frailty; <0.21 defined non-frailty; and ≥0.21 defined frailty [28]. To explore the effect of PA intensity on FI in older adults of different ages, the odds ratio and 95% confidence intervals from the results of the logistic regression were used to compare the significance of the relationship. All data were collected and statistically analyzed using Stata/SE 17.0, with the statistical significance level set at 0.05.

## Results

Table 1 presents a descriptive analysis of the respondents for demographic and health characteristics categorized by vigorous, moderate, and light PA intensities. The results show significant differences in demographic characteristics in all categories except smoking and gender. Fifty-eight percent of the respondents (n = 7162) are categorized in the vigorous PA intensity group, with thirty-one percent (n = 3848) and eleven percent (n = 1238) in the moderate and light PA intensity groups, respectively. Furthermore, one-way ANOVA is employed to compare the FI and metabolic equivalents among the three PA intensity groups. The vigorous-PA intensity group (METs = 6439.6±1885.1) exhibits the lowest mean FI value of 0.237, while the moderate-PA (METs = 1767.4±462.7) and light-PA (METs = 398.0±137.9) intensity groups show values of 0.239 and 0.254, respectively.

S1 Table exhibits the 63 specific health-related deficits utilized in constructing the FI for this study. S2 Table presents the internal consistency of the FI. A Cronbach alpha coefficient of 0.72 indicates the feasibility of constructing the FI. S3 Table shows the internal consistency of the IPAQ categorized by PA intensity, with Cronbach alpha coefficients of 0.79 for the light-intensity group, 0.87 for the moderate-intensity group, and 0.91 for the vigorous-intensity group, thus ensuring the validity of the questionnaire within each intensity group. Fig 2 visually illustrates the frequency distribution of FI for males and females, alongside the FI score age-based trend comparison. Both males and females display a positively skewed distribution. Females, situated farther from the 0 point in the coordinate system than males, indicate a higher FI score in females compared to males. As age increases, FI scores rise for both genders. Notably, after the age of 60, the average FI scores for females consistently exceed those for males. And this gender disparity in FI scores also tends to widen with age.

Fig 3 illustrates the relationship between the occurrence of frailty risk and PA intensity in elderly males. Comparing ORs adjusted for covariates shows that vigorous PA within all age groups of 60–64 years (OR, 0.968; 95% CI, 0.953 to 0.983), 65–69 years (OR, 0.960; 95% CI, 0.946 to 0.973), 70–74 years (OR, 0.964; 95% CI, 0.947 to 0.981), 75–79 years (OR, 0.957; 95% CI, 0.939 to 0.976), and 80 years and older (OR, 0.938; 95% CI, 0.918 to 0.957) is the optimal intensity for reducing the incidence of frailty. The moderate PA after 70 years old, that is, at the age groups of 70–74 years (OR, 0.972; 95% CI, 0.956 to 0.988), 75–79 years (OR, 0.976; 95% CI, 0.958 to 0.993), and 80 years and older (OR, 0.960; 95% CI, 0.946 to 0.973), shows to be statistically significant in reducing the incidence of frailty, but not at optimal intensity.

Fig 4 illustrates the relationship between the occurrence of frailty risk and PA intensity in elderly females. Comparing ORs adjusted for covariates shows that vigorous PA within all age groups of 60–64 years (OR, 0.981; 95% CI, 0.965 to 0.996), 65–69 years (OR, 0.966; 95% CI, 0.953 to 0.979), 70–74 years (OR, 0.972; 95% CI, 0.956 to 0.987), 75–79 years (OR, 0.976; 95% CI, 0.958 to 0.993), and 80 years and older (OR, 0.961; 95% CI, 0.946 to 0.977) is the optimal

**Table 1. Baseline characteristics and demographics of respondents.**

| | Light-PA (n = 1,238) | Moderate-PA (n = 3,848) | Vigorous-PA (n = 7,162) | F or $\chi^2$ | p-value |
|---|---|---|---|---|---|
| **Age range, n (%)** | | | | 504.20 | <0.01** |
| 60–64 years | 281(7.06) | 1,025(25.75) | 2,675(67.19) | | |
| 64–69 years | 301(8.72) | 1,000(28.99) | 2,149(62.29) | | |
| 70–74 years | 232(10.16) | 782(34.24) | 1,270(55.60) | | |
| 75–79 years | 210(14.24) | 570(38.64) | 695(47.12) | | |
| 80 years and older | 214(20.23) | 471(44.52) | 373(35.26) | | |
| **Gender, n (%)** | | | | 3.40 | 0.18 |
| Male | 521(9.57) | 1,707(31.36) | 3,215(59.07) | | |
| Female | 717(10.54) | 2,141(31.47) | 3,947(58.00) | | |
| **Educational background, n (%)** | | | | 65.68 | <0.01** |
| Primary and below | 824(10.16) | 2,436(30.05) | 4,847(59.79) | | |
| Middle school | 258(10.56) | 823(33.69) | 1,362(55.75) | | |
| High school | 117(8.86) | 410(31.04) | 794(60.11) | | |
| College and above | 39(10.34) | 179(47.48) | 159(42.18) | | |
| **Place of residence, n (%)** | | | | 221.43 | <0.01** |
| Urban | 346(10.71) | 1,182(36.57) | 1,704(52.72) | | |
| Urban-rural integration zone | 108(11.89) | 413(45.48) | 387(42.62) | | |
| Town | 522(9.38) | 1,609(28.91) | 3,434(61.71) | | |
| Rural | 183(7.20) | 466(18.32) | 1,894(74.48) | | |
| **Marital Status, n (%)** | | | | 86.09 | <0.01** |
| Married | 829(9.06) | 2,804(30.63) | 5,521(60.31) | | |
| Divorced | 61(11.89) | 140(27.29) | 312(60.82) | | |
| Widowed | 337(13.49) | 882(35.29) | 1,280(51.22) | | |
| Never Married | 11(13.41) | 22(26.83) | 49(59.76) | | |
| **Smoking, n (%)** | | | | 0.52 | 0.77 |
| Yes | 121(10.48) | 353(30.56) | 681(58.96) | | |
| No | 1,117(10.07) | 3,495(31.51) | 6,481(58.42) | | |
| **Alcohol, n (%)** | | | | 55.57 | <0.01** |
| Yes, more than once a month | 209(7.74) | 768(28.42) | 1,725(63.84) | | |
| Yes, but less than once a month | 67(8.27) | 247(30.49) | 496(61.23) | | |
| No | 962(11.01) | 2,833(32.44) | 4,939(56.55) | | |
| **Frailty index, mean(sd)** | 0.254(0.111) | 0.239(0.105) | 0.237(0.101) | 14.25 | <0.01** |
| **METs-minutes/week, mean(sd)** | 398.0(137.9) | 1,767.4(462.7) | 6,439.6(1,885.1) | 4,732.38 | <0.01** |

Percentages may not be total 100% because of rounding. Light-PA = light intensity physical activity group, Moderate-PA = moderate intensity physical activity group, Vigorous-PA = vigorous intensity physical activity group, N = sample size, SD = standard deviation

intensity for reducing the incidence of frailty. Beyond 80 years of age (OR, 0.975; 95% CI, 0.955 to 0.995), moderate PA exhibits a beneficial effect in reducing frailty incidence. Light PA has no effect on reducing the incidence of frailty at any age group for either males or females.

## Discussion

To the best of our knowledge, this is the first study targeting Chinese older adults to compare PA intensity with changes in frailty risk across genders and age groups. The present study found that regardless of whether males or females, the optimal effect of vigorous-intensity PA in reducing the risk of frailty is consistently observed throughout the entire old age career.

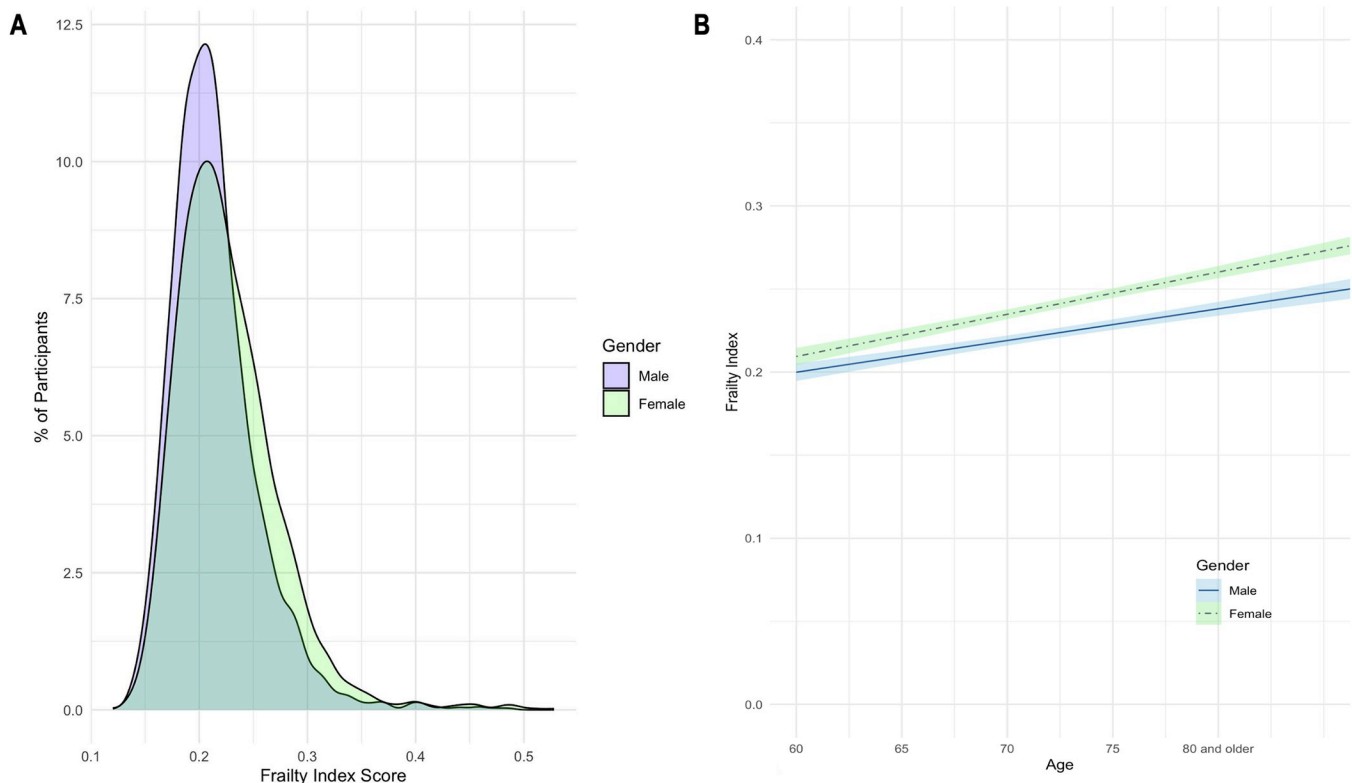

**Fig 2.** A: Distribution of frailty index score for males and females; B: Trend comparison in age-based frailty index score for males and females.

Moreover, the age groups at which moderate-intensity PA reduces the risk of frailty were from age 70 for males and from age 80 for females. And light-intensity PA had no effect on reducing the risk of frailty.

The high percentage of physical activity with vigorous intensity observed in this study reflects a common trend among the elderly population in China at this stage. According to CHARLS survey data up to 2018, the elderly participants in this study were born before the 1960s. This generation experienced significant social and economic transformations in China, which may be one of the reasons they still maintain a high level of physical activity intensity even after retirement. On one hand, China initiated its reform and opening-up policy in 1978. This generation was a major part of the labor force during this period of rapid economic growth, developing strong labor habits in the process [29]. On the other hand, around the 1970s, China's economic structure was primarily based on agriculture and light industry. Due to the low level of mechanization at that time, many tasks still required manual labor [30]. These combined factors have led to the persistence of high physical activity levels among this generation, even after retirement, driven by the labor habits formed in their youth. This phenomenon might differ from patterns observed in other countries.

Notably, older adults of all genders and age groups were able to reduce the risk of frailty with vigorous-intensity PA, which was also supported by previous research. For instance, several epidemiological studies have reported that larger doses of more intense activity may provide additional benefits for cardioprotection [31, 32]. A meta-analysis also demonstrated a significant inverse association between daily step count and all-cause mortality and cardiovascular disease mortality, with the more, the better [33]. Meanwhile, it has also been shown that the proportion of decrease in all-cause mortality decreases as the duration of exercise

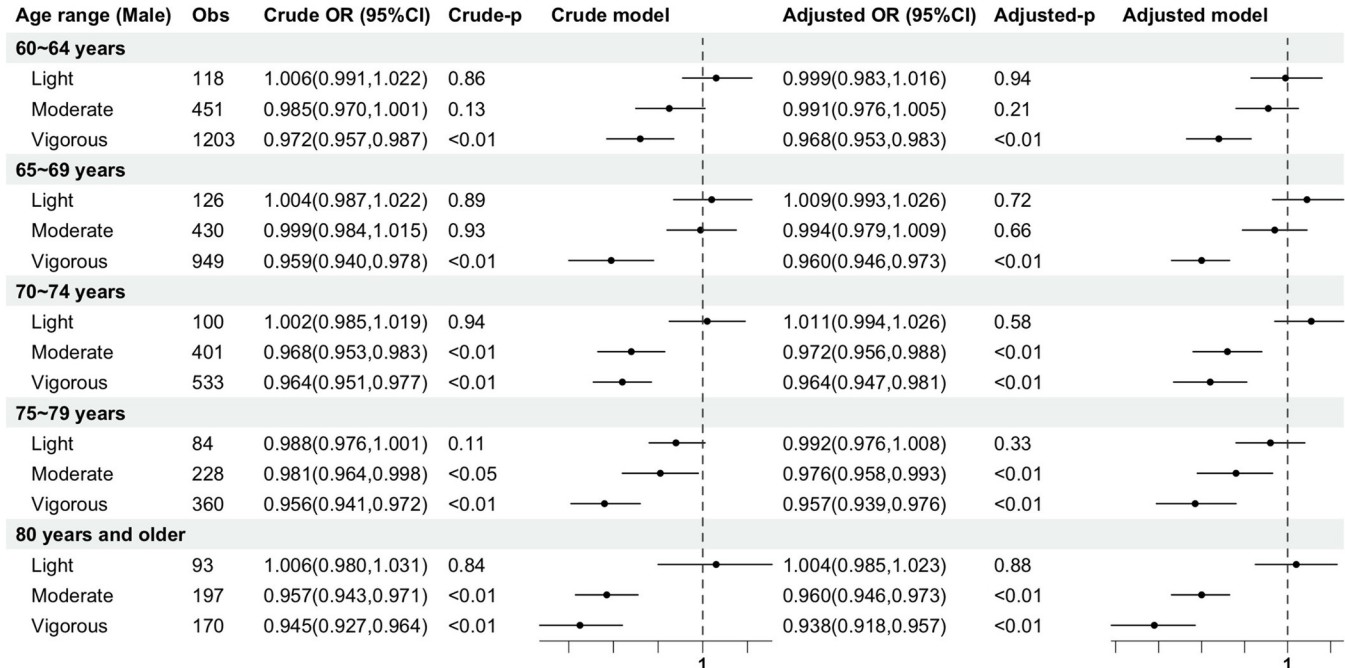

| Age range (Male) | Obs | Crude OR (95%CI) | Crude-p | Crude model | Adjusted OR (95%CI) | Adjusted-p | Adjusted model |
|---|---|---|---|---|---|---|---|
| **60~64 years** | | | | | | | |
| Light | 118 | 1.006(0.991,1.022) | 0.86 | | 0.999(0.983,1.016) | 0.94 | |
| Moderate | 451 | 0.985(0.970,1.001) | 0.13 | | 0.991(0.976,1.005) | 0.21 | |
| Vigorous | 1203 | 0.972(0.957,0.987) | <0.01 | | 0.968(0.953,0.983) | <0.01 | |
| **65~69 years** | | | | | | | |
| Light | 126 | 1.004(0.987,1.022) | 0.89 | | 1.009(0.993,1.026) | 0.72 | |
| Moderate | 430 | 0.999(0.984,1.015) | 0.93 | | 0.994(0.979,1.009) | 0.66 | |
| Vigorous | 949 | 0.959(0.940,0.978) | <0.01 | | 0.960(0.946,0.973) | <0.01 | |
| **70~74 years** | | | | | | | |
| Light | 100 | 1.002(0.985,1.019) | 0.94 | | 1.011(0.994,1.026) | 0.58 | |
| Moderate | 401 | 0.968(0.953,0.983) | <0.01 | | 0.972(0.956,0.988) | <0.01 | |
| Vigorous | 533 | 0.964(0.951,0.977) | <0.01 | | 0.964(0.947,0.981) | <0.01 | |
| **75~79 years** | | | | | | | |
| Light | 84 | 0.988(0.976,1.001) | 0.11 | | 0.992(0.976,1.008) | 0.33 | |
| Moderate | 228 | 0.981(0.964,0.998) | <0.05 | | 0.976(0.958,0.993) | <0.01 | |
| Vigorous | 360 | 0.956(0.941,0.972) | <0.01 | | 0.957(0.939,0.976) | <0.01 | |
| **80 years and older** | | | | | | | |
| Light | 93 | 1.006(0.980,1.031) | 0.84 | | 1.004(0.985,1.023) | 0.88 | |
| Moderate | 197 | 0.957(0.943,0.971) | <0.01 | | 0.960(0.946,0.973) | <0.01 | |
| Vigorous | 170 | 0.945(0.927,0.964) | <0.01 | | 0.938(0.918,0.957) | <0.01 | |

**Fig 3. Comparison in males of the effects of PA intensity on FI across the age group.** Educational background, place of residence, marital status, smoking, and alcohol consumption were added as covariates to the logistic regression in the adjusted model. Light = light-intensity physical activity group; Moderate = moderate-intensity physical activity group; vigorous = vigorous-intensity physical activity group; OR = odds ratio; 95%CI = 95% confidence interval.

| Age range (Female) | Obs | Crude OR (95%CI) | Crude-p | Crude model | Adjusted OR (95%CI) | Adjusted-p | Adjusted model |
|---|---|---|---|---|---|---|---|
| **60~64 years** | | | | | | | |
| Light | 163 | 0.999(0.982,1.016) | 0.94 | | 1.013(0.996,1.030) | 0.12 | |
| Moderate | 574 | 0.996(0.964,1.028) | 0.85 | | 0.999(0.976,1.022) | 0.96 | |
| Vigorous | 1472 | 0.974(0.954,0.994) | <0.05 | | 0.981(0.965,0.996) | <0.05 | |
| **65~69 years** | | | | | | | |
| Light | 175 | 0.990(0.962,1.018) | 0.45 | | 0.992(0.976,1.008) | 0.32 | |
| Moderate | 570 | 0.978(0.954,1.002) | 0.20 | | 0.987(0.968,1.006) | 0.49 | |
| Vigorous | 1200 | 0.977(0.964,0.990) | <0.01 | | 0.966(0.953,0.979) | <0.01 | |
| **70~74 years** | | | | | | | |
| Light | 132 | 1.013(0.994,1.032) | 0.37 | | 1.001(0.983,1.018) | 0.95 | |
| Moderate | 381 | 0.992(0.964,1.020) | 0.76 | | 0.994(0.966,1.022) | 0.73 | |
| Vigorous | 737 | 0.965(0.949,0.981) | <0.01 | | 0.972(0.956,0.987) | <0.01 | |
| **75~79 years** | | | | | | | |
| Light | 126 | 1.009(0.979,1.039) | 0.77 | | 1.004(0.985,1.023) | 0.84 | |
| Moderate | 342 | 0.976(0.956,0.996) | <0.05 | | 0.981(0.961,1.001) | 0.09 | |
| Vigorous | 335 | 0.982(0.965,0.999) | <0.05 | | 0.976(0.958,0.993) | <0.01 | |
| **80 years and older** | | | | | | | |
| Light | 121 | 0.981(0.961,1.001) | 0.10 | | 0.986(0.969,1.004) | 0.14 | |
| Moderate | 274 | 0.977(0.957,0.997) | <0.05 | | 0.975(0.955,0.995) | <0.05 | |
| Vigorous | 203 | 0.959(0.944,0.975) | <0.01 | | 0.961(0.946,0.977) | <0.01 | |

**Fig 4. Comparison in females of the effects of PA intensity on FI across the age group.** Educational background, place of residence, marital status, smoking, and alcohol consumption were added as covariates to the logistic regression in the adjusted model. Light = light-intensity physical activity group; Moderate = moderate-intensity physical activity group; vigorous = vigorous-intensity physical activity group; OR = odds ratio; 95%CI = 95% confidence interval.

increases, but age and exercise dose have not been determined [34]. Distinct from previous studies, the present study identified the optimal intensity of PA required to reduce aging in older adults of all ages and genders through a large sample of observations. These findings enhance our deeper understanding of the relationship between PA intensity and frailty risk in older adults, and they also provide evidence that older adults of different age groups and genders reduce their risk of frailty through appropriate PA intensity.

The identification by different age groups of specific thresholds for reaping the benefits of moderate-intensity PA—specifically, the discrepancy observed between males starting at age 70 and females starting at age 80—is a significant highlight. This variance could stem from inherent physiological distinctions between genders, leading to varying susceptibility to PA intensity and frailty changes—that is, the male-female health-survival paradox.

To date, it has been established that females exhibit a greater prevalence and earlier onset of the aging syndrome associated with cumulative decline in physiological systems [35]. Likewise, FI scores were generally higher in females compared to males [36]. Our results comparing the frequency distributions of FI scores between males and females also supported this trend (Fig 2). However, females with frailty tend to live longer with the syndrome than males [37]. The Seifarth et al. study also showed that, although there were significant differences in gender equality, higher percentages of body mass, and lower levels of PA across age ranges globally, gender-based gaps in life expectancy existed in almost every country for which data existed [38].

This phenomenon is complex and may be related to gender differences in sociology (social roles and resource access across the lifespan) and genetics (hormonal and immunological factors) [39]. From a sociological perspective, these disparities lead to increased male exposure to occupational hazards, unhealthy dietary choices, and elevated alcohol and tobacco consumption—major risk factors for highly fatal non-communicable chronic diseases, ultimately leading to a reduced life expectancy in males [40]. From a genetic perspective, females have two X chromosomes, compared to males longer telomeres, and experience a slower telomere shortening process, potentially contributing to their longer lifespan [41, 42]. The beneficial impact of estrogen on the vascular and lipid profiles of premenopausal females appears to delay and reduce the impact of atherosclerosis. However, the absence of estrogen after menopause may also affect certain diseases in females [41]. Additionally, testosterone has the potential to impede both the inherent and acquired immune responses, leading to a less robust immune system in males. This susceptibility can increase their susceptibility to infections and, ultimately, mortality [43].

Even though we have not established a causal relationship between PA intensity and frailty risk in both sexes, it is a well-known fact that obesity significantly heightens the risk of numerous diseases. And many of the diseases in which FI was constructed to be involved in this study are closely related to obesity. Differences in the distribution of fat and body composition between both sexes across the lifespan, especially in their senior years, may directly contribute to the manifestation of frailty characteristics. Alternatively, they may indirectly influence metabolic alterations, ultimately leading to the development of conditions that heighten the vulnerability to frailty [44, 45]. For instance, males have a tendency to accumulate abdominal fat at an earlier stage compared to females, which results in a more unfavorable metabolic profile [46]. Furthermore, numerous studies indicate that varying levels of PA intensity yield distinct effects in addressing diverse degrees of obesity [47, 48]. Concurrently, PA significantly influences insulin sensitivity [49]. This might be one of the interpretable reasons that elucidates the fluctuations in how different levels of PA intensity influence the incidence rate of frailty among various age groups and genders.

However, it is important to recognize the limitations of this study. The present study relied on older adults' self-reported health status, PA levels, and cross-sectional data from Chinese older adults, which may be subject to social desirability or recall bias. In future studies, the use of precise measurement devices, such as accelerometers, may improve the precision of PA estimates.

In conclusion, the present study demonstrated that vigorous-intensity PA positively influences frailty risk throughout the entire old age career for both males and females. Moderate-intensity PA mitigated frailty risk, commencing at age 70 for males and age 80 for females. Light-intensity PA had no effect on reducing frailty risk. Moderate and vigorous intensity of PA in older adults should be promoted, but guidelines and recommendations must account for optimal associations with PA dose across genders and age groups while also paying special attention to older individuals with preexisting health conditions in order to avoid the potential risks associated with moderate or vigorous intensity of PA.

## Supporting information

**S1 Table. Individual deficits of the frailty index.**
(DOCX)

**S2 Table. Internal consistency of the frailty index.**
(DOCX)

**S3 Table. Internal consistency of the IPAQ (short forms) by PA intensity.**
(DOCX)

## Acknowledgments

The authors would like to thank the investigators in the CHARLS group at Peking University and all respondents.

## Author Contributions

**Conceptualization:** Di Ma, Yulin Sun, Guoyang Chen, Siwei Hao, Zhenping Jiang, Rui Wang, Shuaipeng Hao.

**Data curation:** Di Ma, Siwei Hao.

**Formal analysis:** Di Ma.

**Funding acquisition:** Di Ma, Yulin Sun, Shuaipeng Hao.

**Investigation:** Di Ma, Zhenping Jiang, Rui Wang.

**Methodology:** Di Ma, Yulin Sun, Guoyang Chen.

**Project administration:** Yulin Sun.

**Resources:** Di Ma, Rui Wang, Shuaipeng Hao.

**Software:** Di Ma, Guoyang Chen, Siwei Hao.

**Supervision:** Di Ma, Yulin Sun, Rui Wang.

**Validation:** Di Ma, Zhenping Jiang.

**Visualization:** Di Ma, Shuaipeng Hao.

**Writing – original draft:** Di Ma.

**Writing – review & editing:** Yulin Sun, Guoyang Chen, Siwei Hao, Zhenping Jiang.

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
