## [Decision Letter · Decision Letter 0]

5 Mar 2024

PONE-D-24-02757The association between physical activity intensity and frailty risk among older adults across different age groups and genders: evidence from the four waves survey of CHARLSPLOS ONE

Dear Dr. Sun,

Thank you for submitting your manuscript to PLOS ONE. After careful consideration, we feel that it has merit but does not fully meet PLOS ONE’s publication criteria as it currently stands. Therefore, we invite you to submit a revised version of the manuscript that addresses the points raised during the review process.

We look forward to receiving your revised manuscript.

Kind regards,

Kiyoshi Sanada, PhD

Academic Editor

PLOS ONE

2. We notice that your supplementary tables are included in the manuscript file. Please remove them and upload them with the file type 'Supporting Information'. Please ensure that each Supporting Information file has a legend listed in the manuscript after the references list.

Reviewers' comments:

Reviewer's Responses to Questions

**Comments to the Author**

1. Is the manuscript technically sound, and do the data support the conclusions?

Reviewer #1: Yes

Reviewer #2: Yes

2. Has the statistical analysis been performed appropriately and rigorously? 

Reviewer #1: Yes

Reviewer #2: Yes

3. Have the authors made all data underlying the findings in their manuscript fully available?

Reviewer #1: Yes

Reviewer #2: Yes

4. Is the manuscript presented in an intelligible fashion and written in standard English?

Reviewer #1: Yes

Reviewer #2: Yes

5. Review Comments to the Author

Reviewer #1: I found the study to be very interesting as you presented the results of your research on the relationship between physical activity and frailty according to gender and age group. In this context, I have a few questions, and I would be grateful if you could answer them for me.

1.Are cognitive function tests being conducted, although answers are mainly based on questionnaires such as the iPAQ?

2.Previous studies on the relationship between physical activity and health status at various ages and the relationship between frailty and physical activity have been found in the past. What is the novelty of this study again? Also, please describe and discuss what is different from previous studies.

3.Regarding physical activity, it is believed that it is considerably influenced by the lifestyle of each country. Therefore, we think it would be better to describe the influence of the cultural background of Chinese people.

4.As mentioned in the limitations, the iPAQ questionnaire was used for the physical activity assessment in this study, but would the results be different if an accelerometer or similar instrument were used? I would like to hear your personal opinion.

5.[Comparing ORs adjusted for covariates shows that vigorous PA within all age 190 groups of 60–64 years (OR, 0.968; 95％ CI, 0.953 to 0.983), 65–69 years (OR, 0.960; 95％ CI, 0.946 191 to 0.973), 70–74 years (OR, 0.964; 95％ CI, 0.947 to 0.981), 75–79 years (OR, 0.957; 95％ CI, 0.939 192 to 0.976), and 80 years and older (OR, 0.938; 95％ CI, 0.918 to 0.957) is the optimal intensity for 193 reducing the incidence of frailty. The moderate PA, after 70 years old, that is, at the age groups 194 of 70–74 years (OR, 0.972; 95％ CI, 0.956 to 0.988), 75–79 years (OR, 0.976; 95％ CI, 0.958 to 0.993), 195 and 80 years and older (OR, 0.960; 95％ CI, 0.946 to 0.973), produces a positive effect on reducing 196 the incidence of frailty.]

→We question whether it is appropriate to use language that suggests a causal relationship of positive effects. Since this is a cross-sectional study, we felt that it was somewhat inappropriate.

Reviewer #2: Thank you for your submitting to the PLOS ONE. Although this paper has some original parts, I think it still needs revision before it can be published. Below are the points that caught my attention.

1. The authors state that the coefficient for light-intensity PA is 3.3, the coefficient for moderate-intensity PA is 4.0, and the coefficient for vigorous-intensity PA is 8.0. Usually, low intensity is defined as less than 3 METS, moderate intensity is defined as 3-6 METs, and high intensity is defined as 6 METs or more, but did this affect the results of this study?

2. For logistic regression analysis, at least the items that are found to be significant in Table 1 should be included as covariates.

3. Recent studies have often used objective indicators such as 3-axis accelerometers, but it is necessary to add the validity of classifying physical activity intensity using IPAQ.

4. Recent studies have reported the usefulness of sedentary time and MVPA. The discussion of this study needs to be updated (ex. PMID: 38349064, PMID: 38332942, PMID: 38219269, PMID: 38030967).

167 demographic characteristics in all categories except smoking and gender. Fifty eight percentage of the

6. PLOS authors have the option to publish the peer review history of their article (what does this mean?). If published, this will include your full peer review and any attached files.

Reviewer #1: **Yes: **Soma Tsujishita

Reviewer #2: No

---

## [Author Response · Author response to Decision Letter 0]

19 Mar 2024

We appreciate the efforts and comments of the editor and reviewers on our manuscript, “The association between physical activity intensity and frailty risk among older adults across different age groups and genders: evidence from the four waves survey of CHARLS” (PONE-D-24-02757). The comments were helpful in improving the present manuscript. We modified/revised the manuscript, following the reviewer’s comments, concerns, and suggestions. To facilitate the review of these modifications, we have highlighted all changes in RED in the revised manuscript, which we are resubmitting for your consideration. 

Reviewer #1

Comment 1: Are cognitive function tests being conducted, although answers are mainly based on questionnaires such as the IPAQ?

Response: Thank you for your comments. Regarding this question, the FI framework created by Rockwood and Mitnitski was utilized as the dependent variable in our study, including several categories of health problems negatively associated with increasing age. We selected 20 items from the MMSE and CSI-D scales, used to assess cognitive function in older adults, as components of the FI (Table S1). We also conducted a Cronbach internal consistency test (Cronbach Alpha = 0.73) to ensure the robustness of the items (Table S2). The results of the IPAQ test served as the independent variable in this study, representing physical activity intensity. We aim to investigate whether physical activity reduces the risk of frailty in older adults. If so, we aim to determine the optimal intensity of physical activity for older adults of different genders and age groups.

Comment 2: Previous studies on the relationship between physical activity and health status at various ages and the relationship between frailty and physical activity have been found in the past. What is the novelty of this study again? Also, please describe and discuss what is different from previous studies.

Response: Thank you for your comments. We acknowledge your point of view and recognize that highlighting differences from previous studies can give readers a clearer understanding of the study. Therefore, we have revised and supplemented the original paper in the discussion section (line 235-247).

Comment 3: Regarding physical activity, it is believed that it is considerably influenced by the lifestyle of each country. Therefore, we think it would be better to describe the influence of the cultural background of Chinese people.

Response: Thank you for your comments. Indeed, as you mentioned, older adults in China usually have a preference for practicing Tai Chi in the mornings at parks and taking part in square dancing activities at community squares in the evenings. This category of physical activities is influenced by traditional Chinese culture, setting it apart from those in other countries. We have already supplemented this aspect in the introduction section (line 65-71).

Comment 4: As mentioned in the limitations, the iPAQ questionnaire was used for the physical activity assessment in this study, but would the results be different if an accelerometer or similar instrument were used? I would like to hear your personal opinion.

Response: Thank you for your comments. We fully acknowledge the advantage of objective measurement over questionnaire-based assessments in providing accuracy. The use of the IPAQ questionnaire for evaluating physical activity may introduce biases due to its reliance on subjective recollection and estimation. Conversely, accelerometers or similar instruments offer objective, real-time activity data, and there have been studies confirming that the questionnaire might be higher than the objective measurements. However, due to financial constraints, large-scale objective measurements are currently not feasible for us. When utilizing questionnaire data, we first subjected the questionnaire results to consistency testing to ensure a certain level of validity. Moreover, our data stems from a national survey, and with a sufficiently large sample size, we believe the results hold a degree of persuasiveness. In the future, we intend to explore the use of accelerometers or similar objective measurement methods on a small-scale for comparative analysis with questionnaires while ensuring randomization to further enhance the credibility and accuracy of the findings.

Comment 5: [Comparing ORs adjusted for covariates shows that vigorous PA within all age 190 groups of 60–64 years (OR, 0.968; 95％ CI, 0.953 to 0.983), 65–69 years (OR, 0.960; 95％ CI, 0.946 191 to 0.973), 70–74 years (OR, 0.964; 95％ CI, 0.947 to 0.981), 75–79 years (OR, 0.957; 95％ CI, 0.939 192 to 0.976), and 80 years and older (OR, 0.938; 95％ CI, 0.918 to 0.957) is the optimal intensity for 193 reducing the incidence of frailty. The moderate PA, after 70 years old, that is, at the age groups 194 of 70–74 years (OR, 0.972; 95％ CI, 0.956 to 0.988), 75–79 years (OR, 0.976; 95％ CI, 0.958 to 0.993), 195 and 80 years and older (OR, 0.960; 95％ CI, 0.946 to 0.973), produces a positive effect on reducing 196 the incidence of frailty.]

→We question whether it is appropriate to use language that suggests a causal relationship of positive effects. Since this is a cross-sectional study, we felt that it was somewhat inappropriate.

Response: Thank you for your comments, and we strongly agree with your comments. We recognize that this is an observational study using cross-sectional data and cannot prove the existence of causal relationships or causal directions. We have corrected language that is prone to ambiguity (line 201-204).

Reviewer #2

Comment 1: The authors state that the coefficient for light-intensity PA is 3.3, the coefficient for moderate-intensity PA is 4.0, and the coefficient for vigorous-intensity PA is 8.0. Usually, low intensity is defined as less than 3 METS, moderate intensity is defined as 3-6 METs, and high intensity is defined as 6 METs or more, but did this affect the results of this study?

Response: Thank you for your comments. After revisiting the section of the manuscript regarding the definition of PA intensity coefficients and reviewing additional literature, we have confirmed that the utilization of coefficients 3.3, 4.0, and 8.0 to represent light, moderate, and vigorous intensity, respectively, aligns with the guidelines outlined in the International Physical Activity Questionnaire Data Processing and Analysis Guidelines. Additionally, we observed another part of the studies that was consistent with the categorization criteria proposed by the reviewers. Based on the current findings regarding the validity of the IPAQ, it appears that the IPAQ demonstrates better validity in large-scale surveillance studies comparing groups within or between countries rather than at the individual level. When given an adequate sample size, we have reason to believe that the choice of PA intensity subgroup definition would have a negligible impact on the research outcomes.

Comment 2: For logistic regression analysis, at least the items that are found to be significant in Table 1 should be included as covariates.

Response: Thank you for your comments. Educational background, place of residence, marital status, smoking, and alcohol consumption that were significant in Table 1 were added as covariates in the logistic regression of the adjusted model, whereas no covariates were added in the logistic regression of the crude model. We have corrected the content of the legend (Fig 3 and Fig 4) and elaborated on it.

Comment 3: Recent studies have often used objective indicators such as 3-axis accelerometers, but it is necessary to add the validity of classifying physical activity intensity using IPAQ.

Response: Thank you for your comments. We deeply appreciate the time and expertise you have invested in reviewing our manuscript, and we recognize the importance of validity when classifying PA intensity using questionnaire measures. Our study using data from the CHARLS national database resulted in our inability to analyze it using the Test-Retest method, the Parallel Form Test method (comparison with objective measuring tools), and others. We ultimately analyzed the internal consistency of the IPAQ questions within the light, moderate, and vigorous intensity groups using the Cronbach coefficients to ensure the validity of the PA intensity subgroups and have added content to this analysis in the Statistical Methods section (line 155-158), the Results section (line 186-189), and the Supporting Information section (Table S3) of the manuscript.

Comment 4: Recent studies have reported the usefulness of sedentary time and MVPA. The discussion of this study needs to be updated (ex. PMID: 38349064, PMID: 38332942, PMID: 38219269, PMID: 38030967).

Response: Thank you for your comments. We carefully read the latest research developments you provided and reexamined our discussion section. The Mendelian randomization study results have provided new evidence for the causal relationship and direction between PA and reduced frailty risk. Consequently, we have removed the content regarding uncertain causal relationships from the limitation (line 290-294). [Nonetheless, even recognizing that the Biological Gradient (dose-response relationship) is only one component of Hill's Criteria for Causation, we have taken care to avoid asserting any causal relationships and have instead restricted our discussion to the relationship between PA intensity and frailty risk among different age groups and genders.]

---

## [Decision Letter · Decision Letter 1]

2 May 2024

PONE-D-24-02757R1The association between physical activity intensity and frailty risk among older adults across different age groups and genders: evidence from the four waves survey of CHARLSPLOS ONE

Dear Dr. Sun,

Thank you for submitting your manuscript to PLOS ONE. After careful consideration, we feel that it has merit but does not fully meet PLOS ONE’s publication criteria as it currently stands. Therefore, we invite you to submit a revised version of the manuscript that addresses the points raised during the review process.

We look forward to receiving your revised manuscript.

Kind regards,

Kiyoshi Sanada, PhD

Academic Editor

PLOS ONE

Reviewers' comments:

Reviewer's Responses to Questions

**Comments to the Author**

1. If the authors have adequately addressed your comments raised in a previous round of review and you feel that this manuscript is now acceptable for publication, you may indicate that here to bypass the “Comments to the Author” section, enter your conflict of interest statement in the “Confidential to Editor” section, and submit your "Accept" recommendation.

Reviewer #3: (No Response)

2. Is the manuscript technically sound, and do the data support the conclusions?

Reviewer #3: Yes

3. Has the statistical analysis been performed appropriately and rigorously? 

Reviewer #3: Yes

4. Have the authors made all data underlying the findings in their manuscript fully available?

Reviewer #3: Yes

5. Is the manuscript presented in an intelligible fashion and written in standard English?

Reviewer #3: Yes

6. Review Comments to the Author

Reviewer #3: In this manuscript, the authors investigated the relationship between physical activity intensity and frailty risk in different age groups and genders. This study showed that vigorous intensity physical activity decreases the risk of frailty, regardless of gender or age group. Additionally, moderate intensity physical activity reduces the risk of frailty after the 70s in men and after the 80s in women.

This manuscript may be potentially interesting and clinical significance, however, there are several key concerns that need to be addressed.

Comments:

1. Considering that the novelty of this study is its analysis by age group, we feel that it would be more significant to identify changes from middle age to older age. Is the sample size a reason for limiting the subjects to those 60 years of age or older?

2. Is there a difference in frailty index by age groups? Age-related changes in the frailty index seem to be important basic information.

3. It seems to me that a very high percentage of the elderly have a high intensity of physical activity, does this reflect the general situation of the elderly? Or is this a characteristic trend of the Chinese? Please discuss it.

4. Is the intensity of physical activity defined as high intensity in this study considered comparable to the intensity defined in previous studies?

5. Do the results of this study strongly reflect the social background and culture of China? If so, please add “Chinese older adults” somewhere in the title.

7. PLOS authors have the option to publish the peer review history of their article (what does this mean?). If published, this will include your full peer review and any attached files.

Reviewer #3: **Yes: **Natsuki Hasegawa

---

## [Author Response · Author response to Decision Letter 1]

25 May 2024

We again appreciate the efforts and comments of the editor and reviewers on our manuscript during the second review process, “The association between physical activity intensity and frailty risk among older adults across different age groups and genders: evidence from the four-wave survey of CHARLS” (PONE-D-24-02757R1). These comments were helpful in improving the present manuscript. We modified/revised the manuscript, following the reviewer’s comments, concerns, and suggestions. To facilitate the review of these modifications, we have highlighted all changes in RED in the revised manuscript, which we are resubmitting for your consideration. 

Reviewer #3

Comment 1: Considering that the novelty of this study is its analysis by age group, we feel that it would be more significant to identify changes from middle age to older age. Is the sample size a reason for limiting the subjects to those 60 years of age or older?

Response: Thank you for your professional comments in reviewing our manuscript. In the initial stages, our ideas coincided with your comments. Due to the baseline age for the CHARLS database starting at 45 years old, we initially attempted to include participants from these age ranges and different genders. This would more accurately reflect changes in an individual's frailty index from middle age to older age and the effects of physical activity on its improvement. However, after dividing the sample into subgroups by age and gender, we found that the sample size was insufficient to support reliable results. Thus, this will also become a key focus of our future work as future waves of the survey increase the sample size.

Comment 2: Is there a difference in frailty index by age groups? Age-related changes in the frailty index seem to be important basic information.

Response: Thank you for your professional and pertinent recommendations. We acknowledge your point of view and recognize that age-related trends in the frailty index are indeed important basic information in the present research. Therefore, we have added a comparative graph showing frailty index trends for males and females aged 60 to 80 and older in Figure 2B and included this in the descriptive statistics section of the results (lines 191-196).

Comment 3: It seems to me that a very high percentage of the elderly have a high intensity of physical activity, does this reflect the general situation of the elderly? Or is this a characteristic trend of the Chinese? Please discuss it.

Response: Thank you for your professional comments. Regarding the high percentage of high-intensity PA presented in this study, which indeed reflects a characteristic trend of Chinese elderly at the current stage. This might be related to the changes in China's social background and economy in recent decades. We have discussed it in terms of the participants' age structure, China's reform policies, and the country's economic structure (lines 240-252).

The discussion in the main text follows: “The high percentage of physical activity with vigorous intensity observed in this study reflects a common trend among the elderly population in China at this stage. According to CHARLS survey data up to 2018, the elderly participants in this study were born before the 1960s. This generation experienced significant social and economic transformations in China, which may be one of the reasons they still maintain a high level of physical activity intensity even after retirement. On one hand, China initiated its reform and opening-up policy in 1978. This generation was a major part of the labor force during this period of rapid economic growth, developing strong labor habits in the process [29]. On the other hand, around the 1970s, China’s economic structure was primarily based on agriculture and light industry. Due to the low level of mechanization at that time, many tasks still required manual labor [30]. These combined factors have led to the persistence of high physical activity levels among this generation, even after retirement, driven by the labor habits formed in their youth. This phenomenon might differ from patterns observed in other countries.”

Comment 4: Is the intensity of physical activity defined as high intensity in this study considered comparable to the intensity defined in previous studies?

Response: Thank you for your professional comments. Regarding the question of the definition of high-intensity PA (or vigorous PA) in our study, we conducted an extensive literature review and noted that there are indeed variations in how PA intensity is defined across different countries and versions of IPAQ translated from English. However, we found that most researchers utilizing the CHARLS database align with our study's definition of PA intensity.

For instance, in a study investigating the relationship between PA and depression, using IPAQ-SF questionnaire defined coefficients for low, moderate, and high PA intensity as 3.3, 4.0, and 8.0, respectively (Materials and Methods section) [1]. Similarly, another study on the relationship between PA and intrinsic capacity employed the same coefficients (Materials and Methods section) [2]. Additionally, research focusing on the impact of PA on daily physical function utilized consistent PA intensity standards (Materials and Methods section) [3]. Furthermore, a survey investigating physical inactivity among older Chinese adults used same coefficients to define PA intensity (Measurements section) [4]. Moreover, a study on the influence of PA on cognitive function in Chinese diabetes patients also employed a similar method to define PA score (Measurement of physical activity section) [5].

The reason for the existence of ambiguities may stem from differences in using database types and translations of IPAQ questionnaires from English into different languages. Additionally, we conducted an internal consistency test for PA intensity within subgroups, with the alpha coefficient indicating a certain level of reliability in our study of PA intensity classification (Table S3, Cronbach Alpha in groups of Light, Moderate, Vigorous, and Total are 0.79, 0.87, 0.91 and 0.85, respectively). Hence, the definition of high intensity in this study is unlikely to have influenced the robustness of the analysis of outcomes, despite the lack of a comprehensive test that includes validity testing due to the limitation.

Reference:

1. Jin X, Liu H, Niyomsilp E. The Impact of Physical Activity on Depressive Symptoms among Urban and Rural Older Adults: Empirical Study Based on the 2018 CHARLS Database. Behavioral Sciences. 2023 Oct 21;13(10):864.

2. Zhou M, Kuang L, Hu N. The Association between Physical Activity and Intrinsic Capacity in Chinese Older Adults and Its Connection to Primary Care: China Health and Retirement Longitudinal Study (CHARLS). International Journal of Environmental Research and Public Health. 2023 Mar 31;20(7):5361.

3. Tian Y, Shi Z. Effects of physical activity on daily physical function in Chinese middle-aged and older adults: A longitudinal study from CHARLS. Journal of Clinical Medicine. 2022 Nov 2;11(21):6514.

4. Li X, Zhang W, Zhang W, Tao K, Ni W, Wang K, Li Z, Liu Q, Lin J. Level of physical activity among middle-aged and older Chinese people: evidence from the China health and retirement longitudinal study. BMC Public Health. 2020 Dec;20:1-3.

5. Bai A, Tao L, Huang J, Tao J, Liu J. Effects of physical activity on cognitive function among patients with diabetes in China: a nationally longitudinal study. BMC Public Health. 2021 Dec;21:1-9.

Comment 5: Do the results of this study strongly reflect the social background and culture of China? If so, please add “Chinese older adults” somewhere in the title.

Response: Thank you once again for your insightful comments and suggestions. We have carefully considered it and realized that the results of our study do actually reflect the social background and culture of China. Therefore, we expanded the database abbreviation CHARLS in title to its full name, the China Health and Retirement Longitudinal Survey, to ensure accuracy in terminology use and scope of application of regional findings. The corrected title is "The association between physical activity intensity and frailty risk among older adults across different age groups and genders: evidence from four waves of the China Health and Retirement Longitudinal Survey".

---

## [Decision Letter · Decision Letter 2]

30 May 2024

The association between physical activity intensity and frailty risk among older adults across different age groups and genders: evidence from four waves of the China Health and Retirement Longitudinal Survey

PONE-D-24-02757R2

Dear Dr. Sun,

We’re pleased to inform you that your manuscript has been judged scientifically suitable for publication and will be formally accepted for publication once it meets all outstanding technical requirements.

Kind regards,

Kiyoshi Sanada, PhD

Academic Editor

PLOS ONE

Additional Editor Comments (optional):

Reviewers' comments:

Reviewer's Responses to Questions

**Comments to the Author**

1. If the authors have adequately addressed your comments raised in a previous round of review and you feel that this manuscript is now acceptable for publication, you may indicate that here to bypass the “Comments to the Author” section, enter your conflict of interest statement in the “Confidential to Editor” section, and submit your "Accept" recommendation.

Reviewer #1: All comments have been addressed

Reviewer #2: All comments have been addressed

Reviewer #3: All comments have been addressed

2. Is the manuscript technically sound, and do the data support the conclusions?

Reviewer #1: Yes

Reviewer #2: Yes

Reviewer #3: Yes

3. Has the statistical analysis been performed appropriately and rigorously? 

Reviewer #1: Yes

Reviewer #2: Yes

Reviewer #3: Yes

4. Have the authors made all data underlying the findings in their manuscript fully available?

Reviewer #1: Yes

Reviewer #2: Yes

Reviewer #3: Yes

5. Is the manuscript presented in an intelligible fashion and written in standard English?

Reviewer #1: Yes

Reviewer #2: Yes

Reviewer #3: Yes

6. Review Comments to the Author

Reviewer #1: (No Response)

Reviewer #2: All comments 1 to 4 have been answered. I am fully satisfied with this, so I am giving permission to publish it.

Thank you for submitting in PLOS ONE.

Reviewer #3: (No Response)

7. PLOS authors have the option to publish the peer review history of their article (what does this mean?). If published, this will include your full peer review and any attached files.

Reviewer #1: No

Reviewer #2: No

Reviewer #3: **Yes: **Natsuki Hasegawa

---

## [Editor Report · Acceptance letter]

31 May 2024

PONE-D-24-02757R2 

PLOS ONE

Dear Dr. Sun, 

I'm pleased to inform you that your manuscript has been deemed suitable for publication in PLOS ONE. Congratulations! Your manuscript is now being handed over to our production team.

Kind regards, 

on behalf of

Dr. Kiyoshi Sanada 

Academic Editor

PLOS ONE